

# Social influence on the effectiveness of virtual fencing in sheep

Danila Marini[1,2], Tellisa Kearton[1,2], Jackie Ouzman[3], Rick Llewellyn[3], Sue Belson[2] and Caroline Lee[1,2]

[1] School of Environmental and Rural Science, University of New England, Armidale, New South Wales, Australia
[2] Agriculture and Food, Commonwealth Scientific and Industrial Research Organisation, Armidale, New South Wales, Australia
[3] Agriculture and Food, Commonwealth Scientific and Industrial Research Organisation, Adelaide, South Australia, Australia

Corresponding author
Danila Marini,
danila.marini@csiro.au

## ABSTRACT

Early virtual fencing trials have effectively contained small groups of sheep within set areas of a paddock when all animals were wearing manual electronic collars. With sheep farming commonly involving large flocks, a potential cost-effective application of virtual fencing would involve applying equipment to only a proportion of the flock. In this study, we tested the ability of virtual fencing to control a small flock of sheep with differing proportions of the group exposed to the virtual fence (VF). Thirty-six Merino sheep were identified as leaders, middle or followers by moving them through a laneway. The sheep were then allocated to groups balanced for order of movement. The groups ($n = 9$ per group) included applying the VF to the following proportions of animals within each group: (1) 100% ($n = 9$ VF) (2) 66% ($n = 6$ VF; $n = 3$ no VF) (3) 33% ($n = 3$ VF; $n = 6$ no VF) (4) 0% (no VF; free to roam the paddock). The groups were given access to their own paddock (80 × 20 m) for two consecutive days, six hours per day, with the VF groups prevented from entering an exclusion zone that covered 50% of the north side of the paddock. During these hours, VF interactions, behavioural time budgets, and body temperature were recorded as measures of stress, and location was tracked with GPS. Group 100% VF and Control were tested on the first two days and groups 33% VF and 66% VF were tested on the following two days. During VF implementation the 100% VF and 66% VF group were successfully prevented from entering the exclusion zone. Having only 33% of the flock exposed to the virtual fence was not successful, with the sheep pushing forward through the VF to join flock mates in the exclusion zone. For learning to respond to the audio cue, sheep in the 33% group received more electrical stimuli with a 0.51 proportion for the ratio of electrical stimuli to audio cue, compared to 0.22 and 0.28 for the 100% and 66% groups, respectively. There were small differences in behavioural patterns of standing and lying on both days of testing, with the 100% VF and 66% VF groups spending more time lying. Although stress-induced hyperthermia did not occur in any of the VF groups, body temperature differed in the 33% VF group. There were no differences in temperature measures between the control and 100% VF animals. This study demonstrates that for a short period, controlling two-thirds of the flock was equally as effective as virtually fencing all animals, while controlling one-third of a flock with a virtual fence was not effective. For the short term, it appears that implementing the VF to a proportion of the flock

can be an effective method of containment. Due to the limitations of this study, these results warrant further testing with larger flocks and for longer periods.

## INTRODUCTION

Virtual fencing is an emerging technology with the ability to contain livestock within a boundary through the implementation of a warning cue (audio tone) and a negative stimulus (electrical stimulus). Through associative learning animals' pair an audio warning with an electrical stimulus and quickly learn to avoid the virtual fence by responding to the audio warning alone (*Campbell et al., 2018*; *Marini et al., 2018b*). Virtual fencing for cattle currently works on the premise that all animals require the virtual fence to be implemented, however this method may not be economically feasible in the Australian sheep industry due to the large size of many flocks. The efficacy of virtually fencing and controlling only a proportion of sheep in a flock using collars could provide an opportunity to reduce costs.

Currently, there is no commercially available virtual fencing system for sheep. Virtual fencing in sheep has been implemented experimentally using commercially available dog collars but as a new technology, it is important to measure the impact of virtual fencing on sheep welfare, particularly in situations that may affect flock interactions. Initial studies in sheep have shown that they readily learn to respond to an audio cue to avoid an electrical stimulus when trained individually (*Marini et al., 2019*; *Marini et al., 2018a*), a fundamental component required for virtual fencing to be ethically acceptable (*Lee, Colditz & Campbell, 2018*). There have been few studies that have evaluated group responses to the virtual fence and whether a group of sheep could be trained in a paddock environment (*Brunberg et al., 2017*; *Jouven et al., 2012*). These studies found mixed success in containing animals within a paddock. In a controlled grazing study using a manually implemented virtual fence with a small flock of six naïve sheep, (*Marini et al., 2018a*) numerous group interactions were observed where the flock approached the virtual fence line together, but only one individual received the VF cues but all of the animals in the group reacted. This suggests that there are social influences acting within a group when virtual fencing is implemented. Indeed, recently, social facilitation of learning virtual fencing responses were demonstrated in beef cattle (*Keshavarzi et al., in press*).

Sheep flock movements are a continuous process of individual and group decisions that influence how a group reacts and moves (*Ramseyer et al., 2009a*). Sheep flock movements are affected by a variety of factors, including the flocks' activity, size, food availability, age, breed and social relationships within the flock (*Hauschildt & Gerken, 2016*; *Pillot et al., 2010*; *Ramseyer et al., 2009b*; *Taylor et al., 2011*). With the strong flocking behaviour and behaviour synchronisation of sheep (*Hauschildt & Gerken, 2015*; *Hauschildt & Gerken, 2016*), there may be potential to only have a proportion of sheep controlled with the virtual fence and still gain effective control (*Taylor et al., 2011*). However, the proportion

of trained to naïve individuals can impact the response of the flock, as observed by *Jouven et al. (2012)*, where a flock with a larger proportion of sheep without the virtual fence resulted in ineffective containment within the virtual fence.

Behavioural time budgets have been used previously in cattle (*Campbell et al., 2017*; *Campbell et al., 2019*) and sheep (*Marini et al., 2018a*) exposed to virtual fencing, as indicators of impact to welfare through behavioural change. Another method of measuring impact on welfare includes measures of body temperature. Stress induced hyperthermia as measured by an increased body temperature has been previously shown to be an indicator of stress and anxiety states in sheep (*Pedernera-Romano et al., 2010*; *Sanger et al., 2011*). These measures will be used to determine the welfare of the animals in the different treatment groups. (*Hauschildt & Gerken, 2015*; *Hauschildt & Gerken, 2016*; *Taylor et al., 2011*).

In this study we tested the ability for virtual fencing to control a small flock of sheep when differing proportions of the group were exposed to the virtual fence (VF). The hypotheses being tested are (1) the virtual fence would be effective at containing a small flock when 66% of animals had an active virtual fence (2) the flock with 66% of animals exposed to the virtual fence would be more effective than 33% exposed and (3) we predict sheep in the 33% group would have altered behavioural time budgets and increased body temperature, but not the 66% and 100%, compared to control sheep. The aim of this study was to determine if virtually fencing differing proportions of sheep in a flock would affect the efficacy of containment within a virtual fence, as well as determine effects on sheep behaviour and body temperature as indicators of animal welfare.

## MATERIALS & METHODS

### Ethical statement

The protocol and conduct of the study were approved by the CSIRO McMaster Laboratory Animal Ethics Committee under the New South Wales Animal Research Act 1985 (approval ARA 17/24). The trial was conducted at Waikerie, South Australia, in the Autumn of 2018. Average minimum temperature was 10.5 °C and maximum temperature was 31.5 °C, with windspeeds between 22–33 kmph.

### Animals

Thirty-six Merino ewes (41.6 ± 0.62 kg) were used in the study. The sheep were marked for visual identification from 1 to 36 using wool paint (Dy-Mark, Australia) placed on the back and either side of the barrel. Sheep were shorn the week prior to testing and required the wool around their neck to be reclipped using handheld shears on the first test day. During the day sheep were kept on a forage barley crop which was grown in the trial paddocks; at night they were moved to holding yards and supplemented with hay (approximately 1 kg each). Water was provided ab libitum in the inclusion zone of the experimental paddock and in the holding yards. Sheep were kept and moved within their experimental groups throughout the trial.

## Equipment

The virtual fence was implemented with Garmin dog training equipment. This included a GPS track and train collar (Garmin TT15, Garmin Ltd, Olathe, KS, USA) and GPS hand-held unit (Garmin Alpha 100, Garmin Ltd, Olathe, KS, USA). The collars administer an audible sound (70–80 dB, 2.7 kHz). The electrical stimulus was set to level 4 (36 mA, 20 µs with 16 pulses delivered per s) as determined by previous studies (*Marini et al., 2018b*). Each collar was paired to its own hand-held unit.

The day prior to the first test day, Thermachron iButton temperature loggers (Embedded Data Systems, Lawrenceburg, USA) were inserted into the vagina to measure core body temperature. These were attached to an EAZI-BREED CIDR® (Zoetis, New Jersey, USA). The CIDR's were leached of hormone in a phosphate-buffered saline (PBS) solution prior to use. They were assembled using heat shrink tubing as previously described (*Fisher et al., 2008*; *Lea et al., 2008*). Temperature was recorded at 1 min intervals for the duration of the trial. The loggers were removed at the completion of testing on day two for each group.

At the beginning of each test day the sheep were fitted with HOBO® Pendant G accelerometers, (Onset Computer Corporation, Pocasset, MA, USA) to measure standing and lying behaviours. These were attached to the outside of the left hind leg mid-side on the cannon bone with a bandage and were removed at the end of each test day. The loggers recorded x, y, x coordinates that corresponded to a state of either standing/grazing, walking/running or lying, this was recorded every 2 s between the hours of 9:30 a.m. to 3:30 p.m. The sheep in this study have not previously worn the loggers.

## Test procedures

The sheep were weighed, then identified as leaders, middle or followers prior to group allocation. Group movement order in sheep has been previously identified as stable and a way to categorise a sheep's position in a flock (*Squires & Daws, 1975*), with sheep at the back of the flock having a more stable order (*Doughty & Hinch, 2014*). To identify group order, the sheep were walked calmly down a lane way six times and visualized using a drone. Sheep that were identified in the front and back of the flock four times were classed as leaders and followers respectively, with eight sheep identified in each category. They were then allocated to group balanced for order, it was ensured that at least one sheep from each subgroup (leader, follower and middle) had the virtual fence implemented within each group i.e., in the 33% group one leader, one follower and one middle group animal had the VF implemented. The groups ($n = 9$ per group) were:

1) 100% of sheep with active virtual fence ($n = 9$ virtual fence)
2) 66% of sheep with active virtual fence ($n = 6$ virtual fence; $n = 3$ no virtual fence)
3) 33% of sheep with active virtual fence ($n = 3$ virtual fence; $n = 6$ no virtual fence)
4) 0% of sheep with active virtual fence ($n = 9$ no virtual fence); free to roam the paddock

Each group of sheep were tested in four separate paddocks (80 × 20 m, Fig. 1) that were as similar as possible. Test groups were visually isolated from each other by an opaque fence line using shade cloth. The flocks were given access to a paddock for two consecutive days for six hours per day. Group 100% and control were tested on the first two days and groups 33% and 66% were tested on the following two consecutive days between 8 am and 4 pm.

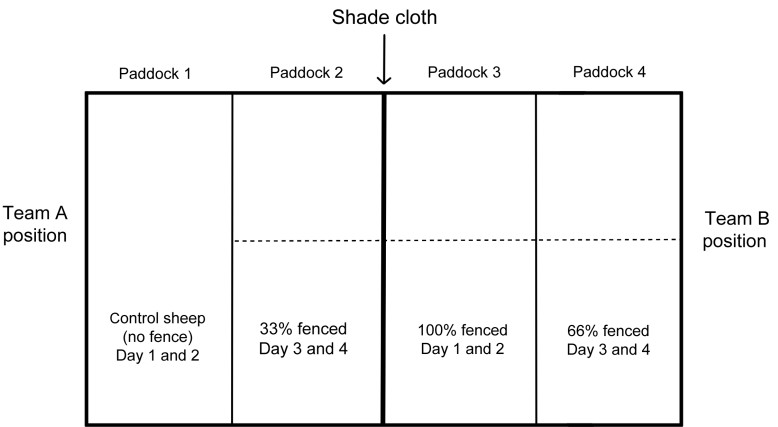

**Figure 1**  Schematic of the experimental setup, the dotted line indicates the virtual fence position.

Sheep in groups with the virtual fence were prevented from entering an exclusion zone that covered 50% of the paddock. Sheep were fitted with Garmin collars at the beginning of each test day. These collars were used to implement the virtual fence and track the sheep through GPS to monitor their location throughout the trial.

### Virtual fence implementation

Prior to testing, a virtual fence and a warning zone (approximately 2 m apart) were discretely marked on the fence lines with coloured tape to ensure the collar operators were spatially consistent at delivering cues. When a sheep entered the warning zone, they were given an audio cue for 2 s. If the sheep continued to walk forward or did not alter their behaviour (i.e., turn around or stop) after the audio cue, an immediate electrical stimulus (<1 s) was given. If the animal ran into the exclusion zone, the audio cues and stimuli were not applied. Once the animal was calm it was again given an audio cue and a stimulus if it proceeded further into the exclusion zone until it either ceased forward movement or began exiting the exclusion zone.

The operators of the Garmin collars were positioned approximately 20 m outside the boundary of the test paddock. Each operator was in control of two sheep using two Garmin handheld GPS units. All sheep were collared but the collars of the uncontrolled sheep were not assigned to a handheld unit. Fence interactions were recorded by the operator using a voice recorder and scribing. All proceedings were videotaped. audio and electrical stimuli delivered were recorded during the trial and were later confirmed using video recordings. Inter-observer agreement was assessed, agreement amongst the observers was 85%.

### Statistics

All data were analysed using the statistical software program R (The R Development Core Team Version 3.5.1.), the *nlme* (*Pinheiro et al., 2018*) package was used for analysis. Data were tested for normality through visual inspection of residual plots and the Shapiro–Wilks test <0.05 was considered statistically significant and $0.1 > P > 0.05$ was considered a statistical tendency. The comparison of the audio cue and electrical stimulus applied to the

groups was analysed using a linear model, data was not normal and required a square root transformation. Analysis of the cues across day within group was tested using a dependent $t$-test. The proportion of electrical stimuli to audio cues received by sheep in the virtual fence groups (number of audio cues was the number of interactions with the virtual fence) was analysed with a binomial proportions test.

Data collected from the hobo loggers were aggregated into 15 min intervals and further collated and presented as the proportion of total time spent in a behaviour for each day. Behaviour was analysed using a linear mixed effects model, fitting group, day and their interaction as a fixed effect, individual sheep was fitted as a random effect. The effects of cohort were accounted by the effect of group.

CircWave software (v 1.4, 2007 R. Hut) was used to perform a cosinor analysis in order to generate a 24 h Fourier curve periodogram for the body temperature data. The transformed fitted body temperature data was subsequently analysed for commonly used body temperature circadian wave parameters, these were: trough, peak, acrophase (the time at which peak occurs), range of oscillation, mesor (mean value of a wave), and amplitude (the difference between the peak and the mesor). Groups were compared with a one-way repeated measure analysis of variance (ANOVA) using a linear mixed effects model with individual sheep as a random effect. Due to data collection occurring across different days, the control group could only be compared with the 100% VF group, and the 33% VF group could only be compared to the 66% VF group for temperature data.

### GPS data

GPS data from Garmin collars was collected for each trial day for each animal. Producing the residency grids, was undertaken in ArcGIS software suite (v9.2; ESRI, Redlands,CA, USA) using a suite of tools from the Hawth's Tools extension (*Beyer, 2004*). The GPS data was interpolated onto a 30 s time step filling missing values with the previous known location, this was done for each animal per day. The trial paddock was divided into a vector grid of 2 m and the individual sheep data was overlaid. For each cell in the grid we summed the time spent and aggregated to produce a residency map for the flock over the trial duration.

## RESULTS

### Effectiveness of the virtual fence

The effectiveness of excluding sheep from the exclusion zone was successful for 100% VF and 66% VF groups but not the 33% VF group (Fig. 2). Fence interactions were recorded for all groups except the control which did not have a fence implemented. Table 1 shows a summary of group interactions with the virtual fence line. There were differences between the groups (100% VF, 66% VF and 33% VF) and the proportion of electrical stimulus to audio cues given ($\chi^2$ (2) = 13.3, $P < 0.05$) over the two days of testing. Proportion for the ratio of electrical stimulus to audio cue were 0.22 for the 100% VF group, 0.28 for the 66% VF group and 0.51 for the 33% VF group, indicating that the 33% VF group received a higher proportion of electrical stimuli. Interactions with the fence increased on day 2 as seen by the increase in audio cues delivered ($F_1$ = 10.7, $P < 0.05$). Two of the sheep

Waikerie virtual fence residency grids

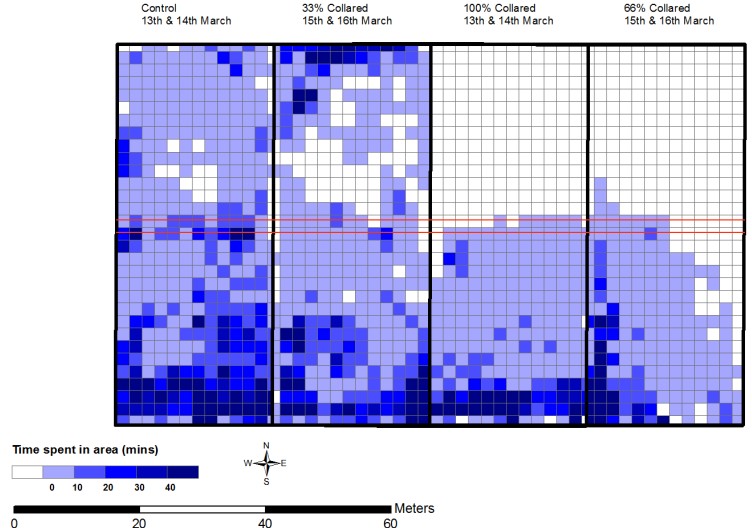

**Figure 2** Residency maps for the flock in each group ($n = 9$ sheep each) at Waikerie showing successful exclusion from northern zone under 100% and 66% active virtual fence groups. Each cell indicates a grid of 2 meters, colour indicates the time spent in each cell (mins). The darker the cell colour, the more time spent.

**Table 1** Summary of total number of group approaches and total number of audio cues and electrical stimuli given to individuals on interaction with the virtual fence. Data presented for groups that had the virtual fence implemented.

| Group (% with fence) | Day 1 | | | Day 2 | | |
|---|---|---|---|---|---|---|
| | Group approaches | Audios | Elec stim | Group approaches | Audios | Elec stim |
| 33 | 4 | 14 | 5 | 9 | 47 | 26 |
| 66 | 3 | 15 | 5 | 5 | 25 | 6 |
| 100 | 12 | 17 | 3 | 12 | 56 | 13 |

with the virtual fence implemented in the 33% VF group reached the maximum number (5) of electrical stimuli allowed to be given to an individual animal within one interaction with the fence. This meant that stimuli were ceased and led to the group crossing over the virtual fence and into the exclusion zone. Once in the exclusion zone the animals were not able to be turned back with the technology this led to the flock residing at the Northern fence line.

## Behaviours

Reactions to the virtual fence cues varied between the groups (Tables 2 and 3). Sheep in the 33% group were the only ones to display negative behaviours in response to the cues, such as running forward in response to application of the audio cue.

**Table 2** Count of behaviours displayed in response to the audio and electrical cues from sheep in the 33%, 66% and 100% virtual fence groups over the two days the virtual fence was implemented.

| Response to audio | 33% | 66% | 100% |
|---|---|---|---|
| Grazing | 17 | 13 | 35 |
| Stop/lifted head | 10 | 9 | 16 |
| Turn | 7 | 10 | 20 |
| Walk forward | 23 | 8 | 2 |
| Run forward | 4 | 0 | 0 |
| Total interactions | 61 | 39 | 73 |

**Table 3** Count of behaviours displayed in response to the audio and electrical cues from sheep in the 33%, 66% and 100% virtual fence groups over the two days the virtual fence was implemented.

| Response to electrical stimulus | 33% | 66% | 100% |
|---|---|---|---|
| No response | 3 | 0 | 0 |
| Turned | 3 | 1 | 7 |
| Jumped and turn | 5 | 9 | 8 |
| Run/jump forward | 5 | 1 | 1 |
| Hop forward | 15 | 0 | 0 |
| Total interactions | 27 | 11 | 15 |

For behavioural time budget, a group and day effect were seen ($P < 0.05$, Fig. 3) for lying ($F_{3,23} = 7.7$) standing ($F_{3,23} = 6$) and walking ($F_{3,23} = 12$). As standing is the opposite of lying, only the results of lying and walking behavior are presented. On day one there was no difference in time spent lying between the control group and the other groups. A difference in lying time was observed between the 100% VF group and the 33% VF group ($t_{32} = -3.17$, $P = 0.003$) with the 33% VF group spending 12% less time lying. On day two, differences in lying were observed between the 66% VF (mean = 28%) compared to the control (mean = 12%, $t_{32} = 4.10$, $P = 0.003$) and 100% VF groups (mean = 13%, $t_{32} = -3.7$, $P = 0.009$). For walking behaviour, the 66% VF group spent significantly less time walking compared to the control and 100% group. On day one the 66% VF group spent 2% of their time walking, the 100% VF group spent 5% ($t_{32} = -3.50$, $P = 0.001$). On day two the differences were observed between 66% VF (mean = 2%) and control (mean = 5%, $t_{32} = -8.64$, $P < 0.05$) and 100% VF (mean = 6%, $t_{32} = -10.61$, $P < 0.05$).

### Temperature

For all the body temperature parameters measured, there were no significant differences between the control and 100% VF group. The 33% v group showed lower values for trough ($F_{(3/30)} = 4.700$, $P = 0.008$) and acrophase ($F_{(3/30)} = 4.434$ $P = 0.011$) than the 66% VF group, and higher values for range of oscillation ($F_{(3/30)} = 5.006$, $P = 0.006$) and amplitude ($F_{(3/30)} = 6.874$, $P = 0.001$) than the 66% VF group (Table 4).
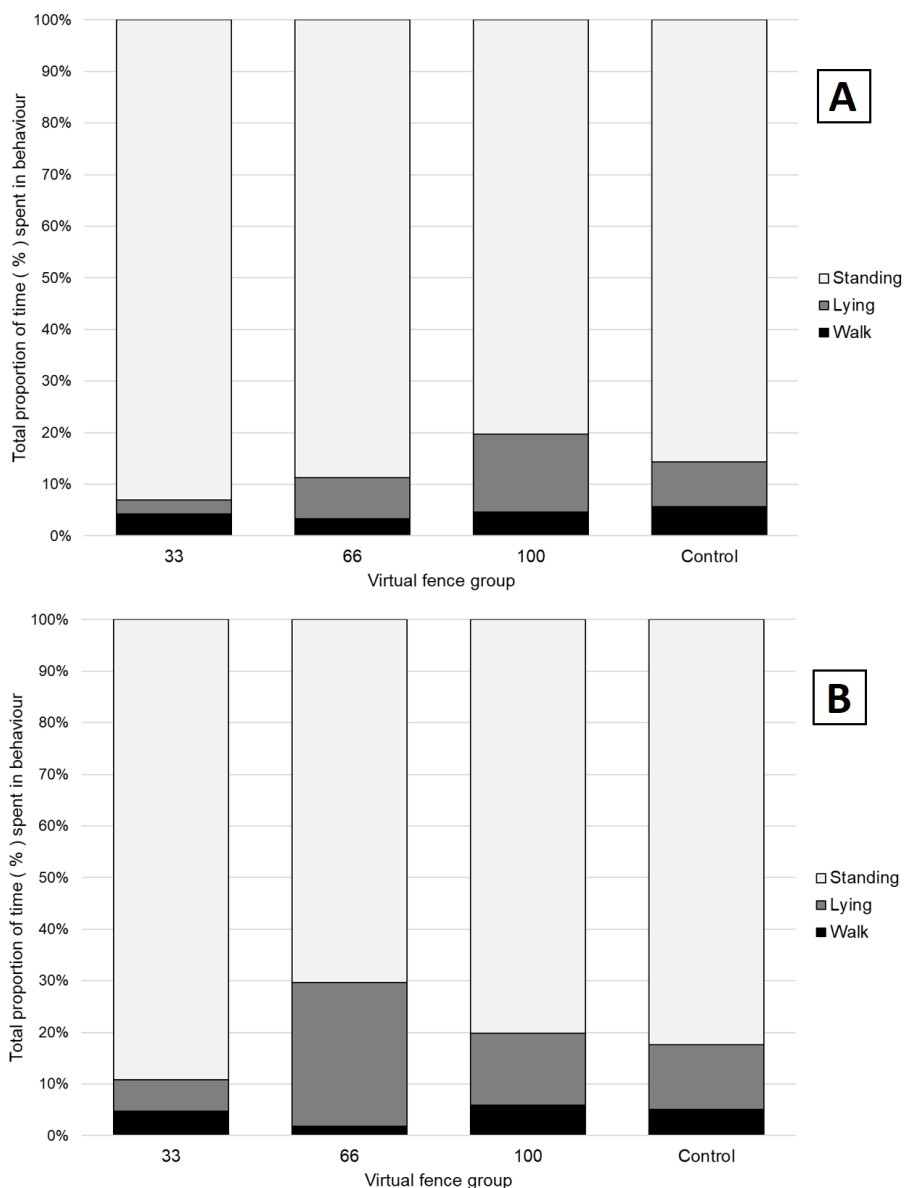

**Figure 3** **Proportion of time spent displaying standing, lying or walking behaviours for each group on (A) day 1 and (B) day 2 of the virtual fencing testing.** The solid black shading is time spent walking, solid grey shading is time spent lying, horizontal lines indicate time spent standing. Time is displayed as a proportion (%) of total activity recorded for each day.

## DISCUSSION

This study looked to determine if application of virtual fencing to differing proportions of sheep in a flock would affect the efficacy of containment during short term deployment. In the context of the current study, it appears that there is a minimum proportion of sheep that require virtual fencing to enable control of the entire group. Implementing the virtual fence on 66% of the flock was highly effective at containing a group of sheep and was similar

**Table 4  Body temperature parameters over the two days of testing for all groups.** Temperature presented as mean degrees °C ±S.E.M.

| | Group | | | | |
|---|---|---|---|---|---|
| | Control | 100% collared | 33% collared | 66% collared | |
| Fitted curve parameters | Mean ± s.e.m. | Mean ± s.e.m. | Mean ± s.e.m. | Mean ± s.e.m. | |
| Mesor | 39.0 ± 0.14 | 39.1 ± 0.12 | 39.0 ± 0.07 | 39.3 ± 0.08 | n.s |
| Peak | 39.5 ± 0.17 | 39.7 ± 0.13 | 39.9 ± 0.07 | 39.9 ± 0.12 | n.s. |
| Trough | 38.4 ± 0.11 | 38.6 ± 0.10 | 38.4 ± 0.11 | 38.8 ± 0.09 | 0.002 |
| Range of oscillation | 1.1 ± 0.09 | 1.1 ± 0.10 | 1.5 ± 0.08 | 1.1 ± 0.09 | 0.001 |
| Amplitude | 0.4 ± 0.04 | 0.6 ± 0.05 | 0.9 ± 0.05 | 0.6 ± 0.09 | 0.006 |
| Acrophase | 17.7 ± 0.78 | 16.2 ± 1.16 | 12.9 ± 1.17 | 15.9 ± 0.17 | 0.03 |
| SD | 0.3 ± 0.02 | 0.3 ± 0.03 | 0.3 ± 0.03 | 0.3 ± 0.02 | n.s. |

to that reported in previous studies with small groups of sheep when 100% of the animals were virtually fenced (*Jouven et al., 2012*; *Marini et al., 2018a*). However, implementing the virtual fence on a smaller proportion of 33% of the flock was not effective at containing the flock within the virtual boundary. Further studies investigating proportions of animals controlled by the virtual fence in larger flock sizes are needed to test these finding over longer periods of time.

We observed some instances where sheep in the 66% VF group, that did not have the virtual fence implemented entered the exclusion zone, but they were encouraged back into the inclusion zone by the movements of the remaining flock. This demonstrates that having trained individuals within a flock can be an effective method of controlling sheep movements. The studies by *Pillot et al. (2010)* and *Taylor et al. (2011)* trained sheep to respond to visual and auditory cues to approach a target (food reward), when these trained sheep were in a flock with naïve animals, they were successful in recruiting the naïve animals to approach the target when the cue was given, thereby influencing the flocks decision and movement patterns. For the animals with VF devices in the 33% VF group, it appeared that their motivation to stay with their 'unfenced' flock mates when they entered the exclusion zone, was greater than the aversiveness of the electrical stimulus of the virtual fence. This was demonstrated by those sheep continuing to walk forward in response to the audio cue and electrical stimulus on several occasions to join their flock mates in the exclusion zone. The results seen in this study are similar to *Jouven et al. (2012)* where, during the 30 min of testing, they found naïve animals crossed over to the exclusion zone often but when they were in the minority they would return to the inclusion zone to re-join the rest of the flock. The precise social dynamics influencing the behaviour of virtually fenced sheep in groups has not yet been explored and future research in this area is needed, particularly for larger flock sizes more commonly found on commercial farms.

These findings should be considered in relation to the limitations of the study. These include the short period of the study (2 days of 6 h per day per treatment), whereas over a longer period, social influences of the uncollared sheep may draw the collared sheep into the exclusion zone and reduce the effectiveness of containment of the sheep. Also, due to limitations on the number of collars and people available to administer the stimuli manually and the number of paddocks available only small group sizes could be tested,
and treatments were not able to be replicated and tested on the same day. This brings in the potential for differences in environmental conditions that may have affected behaviour of the sheep in this study. With the development of automated collars, longer term larger scale replicated studies will be possible, as have been demonstrated in cattle (*Campbell et al., 2019*).

The proportion of audio cue to electrical stimulus is an indicator of sheep learning to respond to the audio cue and avoid receiving the electrical stimulus. The lower the ratio, the better the learning shown by sheep. Within our study, the 33% VF group were the poorest learners with the highest ratio of electrical stimuli to audio cue at 0.50. This ratio was similar to that seen in previous studies where sheep trained individually had a ratio of 0.48 (*Marini et al., 2018b*) and 0.51 (*Marini et al., 2019*). Whereas the 66% VF and 100% VF had a similar mean ratio of 0.28 and 0.22 respectively, these results were more similar to studies that have trained animals in a group, where the mean application of electrical stimuli was 0.19 (*Marini et al., 2018a*). The mean number of electrical stimuli applied in the groups was also significantly different with sheep in the 33% VF group receiving more electrical stimuli over the two days. This has implications for the welfare of the sheep in the 33% VF group as animals may receive high numbers of stimuli, that could negatively impact their welfare. The sheep in the 33% group also displayed some negative behaviour in response to the cues such as running forward on the sound of the audio cue. Each group had an increase in audio cues and electrical stimuli applied on day two, although group approaches only increased slightly. Indicating there may have been more individual interactions with the fence within the group approaches on day two. More approaches were expected on day two as feed availability would have declined in the grazing zone and the higher feed availability in the exclusion zone may have become more attractive. However even with the increased motivation to access fresh pasture, sheep in the 100% VF and 66% VF were still successfully contained by the virtual fence.

The use of manual training collars is labour intensive and limits the number of animals that can be controlled within a group as well as the number of groups able to be tested simultaneously. While, it is possible that the small number of animals used in this study may have impacted the sheep behaviour and flock movements (*Arnold, 1977*; *Taylor et al., 2011*), sheep are able to display normal grazing behaviours in a minimum group of three (*Penning et al., 1993*). Therefore, the study size of nine animals per groups is most likely adequate in relation to expression patterns of behaviour. In the current study the split of animals that did and did not have the virtual fence implemented meant that there were three animals in the 33% VF group that had the virtual fence and three in the 66% VF without, leaving the main portion of the flock containing six animals. It is possible that the larger group of six had more influence over the three animals. In the study by *Jouven et al. (2012)*, they worked with flock sizes of 32 and their largest split of trained animals to naïve was 50:50 leaving them with much larger groups of 16, this would be a big enough group to create a sub flock, which was observed with 13 naïve ewes spending time in their exclusion zone.

In this study, stress induced hyperthermia was not observed, and no significant differences were observed in body temperature parameters for the control and 100%

virtual fenced animals, however small differences were found between 33% and 66% virtual fence animals across four of the seven parameters analysed. Temperatures circadian rhythm is described using several parameters, these are the mesor, peak, trough, range of oscillation, amplitude and acrophase, all of which were used in this study. These measures have been used previously to determine patterns in sheep temperature (*D'Alterio et al., 2012*). The differences observed in these measures of circadian rhythmicity indicate that there was a difference in the pattern of change for body temperature across the day between the 33% VF and 66% VF groups. The results observed in the 66% animals and between the control and the 100% VF group suggest that exposure to the virtual fence and training animals was not inherently stressful, however further investigation is needed to determine whether exposing only a proportion of the flock will negatively impact sheep welfare in the long term. Previous work looking at sheep's initial exposure to the virtual fence electrical stimuli found that the implementation of the electrical stimulus was not highly aversive (*Kearton et al., 2019*). Other work has highlighted the importance of animals learning the audio cue in virtual fencing system in order to maintain good welfare through predictability and controllability (*Lee, Colditz & Campbell, 2018*; *Marini et al., 2019*). However, lack of longer-term trials to date means that the effect of repeated exposure is still to be fully tested with sheep.

Behavioural patterns were different between the groups in this study and differed between the days of testing. Due to design limitations the study was not able to be replicated and the groups required testing on different days adding in additional environmental factors that may have influenced patterns of behaviour. The behaviours of the sheep may also have been affected by the initial adaptation period to the VF. Other studies that have looked at implementing virtual fencing in sheep have not looked at the implementation of the technology over a long period of time and have not been able to look into impacts on normal behaviour (*Brunberg et al., 2017*; *Brunberg, Bøe & Sørheim, 2015*; *Jouven et al., 2012*). In our previous study (*Marini et al., 2018a*), it was found that behavioural time budgets were affected after the virtual fence was removed, through a decrease in lying behaviour, but not during the fence implementation. Studies with cattle have also found that the implementation of the virtual fence had limited impact on behavioural time budgets in cattle (*Campbell et al., 2017*; *Campbell et al., 2019*). The differences in behaviour between the two days of testing and between the groups in this study could have been due to either the short period of time that the sheep spent in the paddocks (which included no acclimation period). Or it could have been due to the restriction and reduction in pasture availability over the time-period with animals in the 100% and 66% group only accessing the inclusion zone. Animals in the control and 33% VF group accessed the entire paddock giving them more opportunity to walk and graze more pasture, which was reflected in the difference in lying time in the 33% group.

## CONCLUSIONS

This study demonstrates that social influences can affect behavioural patterns and the effectiveness of virtual fencing in sheep. Only having a small portion of the flock with an

active virtual fence made the fence ineffective with a much higher proportion of sheep moving forward and entering the exclusion zone, rather than turning in response to the audio cue. There were also indications of potentially negative welfare implications for the animals that were exposed to the virtual fence in the 33% VF group due to the increase proportion in electrical stimulus received. The differences seen in the behavioural patterns may have been due to sheep adapting to the system, as they were only exposed for a short period of time or potential differing environmental conditions due to testing on different days. Further studies using larger group sizes and conducted over longer periods of time are needed to confirm these findings.

## ACKNOWLEDGEMENTS

The experiment was conducted in collaboration with Allen Buckley and Moodie Agronomy from Mallee Sustainable Farming. We thank Damian Mowat and Matthew Williams for their technical assistance.

### Funding

This project is supported by funding from the Australian Government Department of Agriculture, Water and the Environment and CSIRO. The funders had no role in study design, data collection and analysis, decision to publish, or preparation of the manuscript.

### Grant Disclosures

The following grant information was disclosed by the authors:
The Australian Government Department of Agriculture, Water and the Environment and CSIRO.

### Competing Interests

The authors declare there are no competing interests.

### Author Contributions

- Danila Marini conceived and designed the experiments, performed the experiments, analyzed the data, prepared figures and/or tables, authored or reviewed drafts of the paper, and approved the final draft.
- Tellisa Kearton and Sue Belson performed the experiments, analyzed the data, prepared figures and/or tables, authored or reviewed drafts of the paper, and approved the final draft.
- Jackie Ouzman analyzed the data, prepared figures and/or tables, authored or reviewed drafts of the paper, and approved the final draft.
- Rick Llewellyn and Caroline Lee conceived and designed the experiments, performed the experiments, authored or reviewed drafts of the paper, and approved the final draft.

## Animal Ethics

The following information was supplied relating to ethical approvals (i.e., approving body and any reference numbers):

The protocol and conduct of the study were approved by the CSIRO McMaster Laboratory Animal Ethics Committee under the New South Wales Animal Research Act 1985 (approval ARA 17/24).

## Data Availability

Data is available at the CSIRO Data Access Portal: Marini, Danila; Kearton, Tellisa; Ouzman, Jackie; Llewellyn, Rick; Belson, Sue; Lee, Caroline (2020): AEC_17_24_Group effects on Virtual Fencing. v1. CSIRO. Data Collection. https://doi.org/10.25919/5eb3338c32078.

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
