# Peer review of "Social influence on the effectiveness of virtual fencing in sheep"

_PeerJ, doi:10.7717/peerj.10066_

## Round 0.1 · original submission · Major Revisions

Dear authors, the three reviewers had a positive consideration of your work, but they highlighted several concerns that must be addressed at this stage.

Reviewer 1 ·

Basic reporting

The idea is very interesting. I congratulate the authors for a smart idea joining their knowledge in social behavior, learning processes, novel technology for sheep production, and an applied view. However, there is one important limitation. Authors explained that ewes learn from each other, what means that the behavior of each ewe is not independent from the behavior of other ewes, and thus, data from each ewe into each group are not really independent. Some researchers would consider that in this case, the lack of replications implies that the study is not correct, but I disagree with that view (what are real replications in field conditions in which is impossible to really replicate conditions?), so I feel that this does not prevent from publishing it. However, considering the limited of individuals in each condition and the lack of replications, I suggest that the article should be considered as a short communication, as
Another point in which I suggest to be more cautious are the conclusions. Groups were composed by 9 animals, although the main problem is related to big flocks, and there was only one repetition with each percentage of animals. The idea and the study itself are useful, but the conclusions should be more limited.

Experimental design

Limited due to the lack of repetitions.

Validity of the findings

Interesting considerint that are the first approaches to this strategy

Additional comments

The idea is very interesting. I congratulate the authors for a smart idea joining their knowledge in social behavior, learning processes, novel technology for sheep production, and an applied view. However, there is one important limitation. Authors explained that ewes learn from each other, what means that the behavior of each ewe is not independent from the behavior of other ewes, and thus, data from each ewe into each group are not really independent. Some researchers would consider that in this case, the lack of replications implies that the study is not correct, but I disagree with that view (what are real replications in field conditions in which is impossible to really replicate conditions?), so I feel that this does not prevent from publishing it. However, considering the limited of individuals in each condition and the lack of replications, I suggest that the article should be considered as a short communication, as
Another point in which I suggest to be more cautious are the conclusions. Groups were composed by 9 animals, although the main problem is related to big flocks, and there was only one repetition with each percentage of animals. The idea and the study itself are useful, but the conclusions should be more limited.

Introduction
It is a bit long, so I suggest summarizing it approximately 25%. It will also be reorganized, as there are many specific results acknowledged when are not necessary. Including the main concepts would be enough.
I suggest to include first the hypothesis, and then the objective, which responds the hypothesis. The hypothesis was raised by the authors before doing the study, so it should be in past tense.
Considering the authors background, I suggest to include a more directed third hypothesis. Differing is too vague.
M&M
L176-177: Authors stated that “The CIDR’s were leached of hormone in a water bath prior to use.”. However, CIDRs contain progesterone, which is not dissolved in water. Did authors checked that CIDRs were empty from progesterone? If not, how can this affect the treatments?
L202-210: How were the animals fed during those days? The only food were pastures? Therefore, which types and availability. This may affect widely ewes behavior, also in relation to what they are willing to take risks to access or not to other pastures. This. And the pastures on the areas of the paddocks separated by the virtual fences should be described in detail.
L235-236: I am not sure if I understood it correctly, but a “within group” comparison should not be done with a paired t-test, as t-tests are for independent data. If I understood correctly, data were collected more than once from the same animals, and thus, were repeated from the same animals, which is not the same as paired.
L253: Here it is described that “Due to data collection occurring across different days”, but I could not find where that was explained. If it was, I agree that it is an important limitation that should be acknowledged in the Discussion. If data were not recorded simultaneously, many environmental factors could affect the behavior of the animals.
Discussion: I think that this section needs work. Mainly, there is a lot of repetition of results without deep analysis of what is behind those results. I could not have clear the main ideas of some paragraphs. Considering the novelty of the study, I expect authors to go deeper in their findings, but considering the limitations of the study. I hope they can explore more the whole picture instead of each isolated piece (or result).
L311-318: it is mainly a repetition of results. We read the results in the corresponding section, so begin this section with your main new outputs.
L314-316: However, there was not a “control period”, so with few animals and no repetitions, these differences may be related to the flock behavior, and not to the treatments. At least, authors cannot discard this possibility and should mention it.
L319-335: For me it is not clear which is the main point here.
L358-362: I am not sure of this. Does synchronization of social behavior is the same in groups of different size? Are there studies demonstrating this?
L370-400: I suggest going more directly to the point. Why did you recorded behavior and measured temperatures? What did you expect (for that, it would be useful to have a more clear third hypothesis in the Introduction)? Therefore, what are you stating in relation to that hypothesis? As it is here, it seems as if it was measured just to see what is coming and not according to specific hypothesis. The study did not need these data, which are mainly just a complementary data (especially temperature), so, I would expect to have more direct information on what do you consider that is outcoming here (or excluding that data).

·

Basic reporting

A clear level of English is used throughout this article.

The introduction clearly sets the context for the use of virtual fences in sheep, as well as the current state of knowledge on this topic.

The structure of the article agrees with the professional intro/methods/results/discussion structure, and raw data has been shared by the authors.

The article provides relevant results to address the hypotheses raised in the introduction.

Experimental design

The research presented in this article is meaningful to extend and develop virtual fences on a commercial scale. The knowledge gap is clearly presented : can we use virtual fencing at group level ? And can we use virtual fencing effectively by controlling only a proportion of the flock ?

The potential negative impact of virtual fencing on sheep welfare is highlighted, and evaluated by the research presented in this article.

See below for comments on specific parts of the methods/results sections:

158 sqq: were the 36 sheep housed together at night ? And separated into their experimental groups during the day only ? Or were they kept in their experimental groups at all times ?

181: Were the sheep already used to wearing accelerometers ?

Did you consider using heart rate monitors as well as thermometers and accelerometers to evaluate stress and potential negative impact on welfare ?

193: A drone ! How exciting !

198sqq: How did you choose which sheep were going to wear a virtual fence ? I may have missed it, but did you use the leader/follower classification you had established ?

217: Could you give more details on how the impact of the electric shock on the ewes welfare, physiology, behaviour was evaluated, and then deemed acceptable ?

254 : Is there anyway you could balance variations in outdoor temperatures to still be able to compare the variations between all groups ? I am not familiar with temperatures parameters, but it would definitely be very interesting and useful to have a full comparison.

Validity of the findings

All data has been provided by the authors.

Conclusions are clear and linked to the research questions stated in the introduction. This article is a preliminary step into further studies that will need to include larger groups of animals over a larger period of time, which is a limitation clearly stated by the authors.

See below for specific comments.
333: Very interesting indeed.

348: Do you think sheep behaviour would have continued to change with a longer testing phase and the virtual fence might have been ineffective for the 66% group after a week or more ?

360: Good point made on sheep behaviour in small groups

Have you looked at "who" in the flock was wearing the virtual fence ? In the 66% andn 33% groups, which sheep (leader or follower) were equipped with the virtual fence ?
Did you look at the leader/follower status of the sheep withing each group in your analysis ?

Additional comments

Thanks for this very interesting paper, it is always nice to see how virtual fencing is developping as it has a lot of potential for extensively reared livestock.

I hope the comments made above will be clear, I would be particularly interested in knowing more about the link between the personality type of each sheep that you identified (leader/follower) and its impact on the group behaviour, depending on who was wearing the virtual fence.

Reviewer 3 ·

Basic reporting

The topic of the manuscript is very interesting and valuable for the scientific and farming communities. The language is clear and the manuscript is overall well written. The structure follows the classical rules.

Detailed input:
Introduction:
The introduction implies that virtual fencing in general works for sheep. However, there is no virtual fencing system to my knowledge commercially available for sheep and although there is evidence that it worked in some studies, there is also evidence, for the opposite effct (e.g. Brunberg et al. 2015 and 2017). The sequence of the introduction should therefore be improved. After Line 90, it would be important to state the fact from Line 128f that virtual fencing currently does not exist as a commercial system for sheep. Otherwise a false impression is given.

The reference Brunberg et al. (2016) was stated to support the statement that “sheep have shown that they readily learn to respond…”. However, after reading Brunberg’s publication I cannot quite follow your train of thought. Brunberg et al. have stated in their conclusions that “… only nine out of 24 ewes learned to deal with the virtual fencing system in three repetitions…”. It should be noted that for a second experiment containing the ewes, only the nine sheep were chosen from the first experiment which learned to deal with the virtual fence in the Brunberg study.

Line 86f: “Virtual fencing currently requires all animals to have the virtual fence implemented…” This sentence is not correct. Most people assume that this is the case but there is neither a strict rule nor a regulation. Moreover, the described study looks into the feasibility of a reduction of collard animals but no technical changes are targeted. I would recommend to rephrase that sentence.

Line 101: Please clarify the message you want to convey. Did you chose the example of Brunberg et al. (2017) because the lambs were not collared?

Please check manuscript for spaces between numbers and mathematical signs. Sometimes there is a space between the number and the %-sign, sometimes there is no space. The same is true for numbers and =-signs.

References:
E. I. Brunberg, K. E. Bøe & K. M. Sørheim was published 2015 and not 2016, also something regarding the authors’ names seems to be mixed up in the reference.

Experimental design

The research question is well defined and relevant. It is a very intersting topic. However, the study is based on very small numbers. It is correctly mentioned in the manuscript that sheep show a strong flocking and synchronization behaviour (Line 138). That means that for testing the efficacy of virtual fencing, there is really only n = 1 for each treatment (Figure 2) in the described study. Further, the four treatment groups had an allocated paddock. No change of paddock was included in the experimental design as a block factor. In addition, the groups had access to their paddocks for two consecutive days for six hours. Here a better spread across days would have improved the design.
I appreciate that quite a bit if work is involved in such a study. Yet, from the statistical point of view such a small data set with confounding design should be seen as critical.

Detailed input:
Material and Method:
Could you please provide details regarding the weather on the 4 experimental days as this might have an impact on the results as stated by Brunberg et al. (2015). Especially in an international context it can be difficult to guess for the reader what the weather is like in another country.

Could you also please provide information on the amount of wool the sheep had during the trials. Was this shortly after shearing? Have you done specific preparations before the sheep were collared?

Line 231 the use of R packages is mentioned here, then only one package is named. Could you please change it either to singular or add the other packages you have used.

Validity of the findings

The topic is novel and would provide useful information. It is also stated that further research is needed by the authors. However, due to the small numbers included in the study and the difficulties with the study design, the results should be looked at with caution.

The identification of stress signs with a cosinor analysis is a very nice idea. However, as we are talking rather small sample sizes, looking at the results in Table 4 I am not entirely convinced even if some of the results show a significant difference. One should keep in mind that there is a 1 in 20 risk to find a difference that does not exist if we assume α = 0.05. Looking at Figure 3, one can also not identify a clear trend due to the variation, even if the standing behaviour is on both days high for “Treatment 33%”. It might be the case that there is indeed an effect but it does need more data to support it.

On the other hand, Tables 2 and 3 are very interesting to get an understanding about the responses.

Additional comments

It is very difficult to judge the paper as it has positive and negative aspects. I think the experimental design is not strong enough for a full paper. However, there is still valuable information in there if there is a publication option as a technical note. Therefore, it is with regret that I would recommend it for rejection.

---

## Round 0.2 · Minor Revisions

Dear authors, many thanks for tackling the reviewers' suggestions. One of them has minor concerns about the introduction and discussion that may be addressed to some degree.

Reviewer 1 ·

Basic reporting

Authors did a good job, improving the content of the article.I still consider that the Introduction is too long. I suggest to go more directly to what you aim to test. There are too many details describing previous studies.
L80: delete “available”, as it is repeated

Experimental design

The design has an important limitation as different groups were tested in different days. I can understand the limitations in devices and observers, but this should be more strongly considered as a limitation in the Discussion.

Validity of the findings

The findings are important and have potential impact. Are limited due to the limitation of the design, number of animals, etc, but in an area with scarce research, the results might contribute to develop it.

Additional comments

Thank you for your receptiveness.

·

Basic reporting

no comment

Experimental design

Thank you for adressing comments, no further remarks.

Validity of the findings

No comment

Additional comments

Many thanks for this updated version and for adressing reviewer's comments.

Reviewer 3 ·

Basic reporting

The points I have raised in the last review regarding the basic reporting are satisfactorily answered and the manuscript amended accordingly.

Experimental design

The challenges of the very limited experimental design are more prominently reported now. This adds value to the paper and makes sure that readers do take the limitations more sufficiently into account.

Validity of the findings

The above comment also answers this section.

Additional comments

I am ok if the manuscript will get accepted and published now.

---

## Round 0.3 · accepted · Accept

The article is ready for acceptance. Thank you for tackling the reviewers' comments!

Reviewer 1 ·

Basic reporting

The article is ready for acceptance

Experimental design

The article is ready for acceptance

Validity of the findings

The article is ready for acceptance

Additional comments

The article is ready for acceptance